# Organocatalytic atroposelective construction of axially chiral N, N- and N, S-1,2-azoles through novel ring formation approach

Yu Chang[1,2], Chuandong Xie[1,2], Hong Liu[1], Shengli Huang[1], Pengfei Wang[1], Wenling Qin [1✉] & Hailong Yan [1✉]

1,2-Azoles are privileged structures in ligand/catalyst design and widely exist in many important natural products and drugs. In this report, two types of axially chiral 1,2-azoles (naphthyl-isothiazole S-oxides with a stereogenic sulfur center and atropoisomeric naphthyl pyrazoles) are synthesized via modified vinylidene *ortho*-quinone methide intermediates. Diverse products are acquired in satisfying yields and good to excellent enantioselectivities. The vinylidene *ortho*-quinone methide intermediates bearing two hetero atoms at 5-position have been demonstrated as a platform molecule for the atroposelective synthesis of axially chiral 1,2-azoles. This finding not only enrich our knowledge of vinylidene *ortho*-quinone methide chemistry but also provide the easy preparation method for diverse atropisomeric heterobiaryls that were inaccessible by existing methodologies. The obtained chiral naphthyl-isothiazole S-oxides and naphthyl-pyrazoles have demonstrated their potential application in further synthetic transformations and therapeutic agents.

[1] Chongqing Key Laboratory of Natural Product Synthesis and Drug Research, School of Pharmaceutical Sciences, Chongqing University, 401331 Chongqing, P. R. China. [2] These authors contributed equally: Yu Chang, Chuandong Xie. ✉email: wenling.qin@cqu.edu.cn; yhl198151@cqu.edu.cn

**1**,2-Azoles are five-membered heterocycles containing at least one nitrogen atom and privileged structures widely found in pharmaceuticals[1–7], pesticides[8], and chiral catalysts[9] or ligands[10,11] (Fig. 1a). Recently, the atroposelective synthesis of atropisomeric (hetero)biaryl motifs and their stereochemical recognition in biological or chemical targets are hot spots in drug discovery and catalyst/ligand design[12–24]. Therefore, the installation of 1,2-azole moieties into (hetero)biaryl system leads to additional skeletal heterogeneity in atropisomeric architectures. To our surprise, only atropisomeric pyrazole has been prepared through the atroposelective formation of aryl-aryl bond to date (Fig. 1b)[25,26]. The atroposelective procedure for other 1,2-azoles such as isoxazole, isothiazole, and its S-oxides has not been explored probably because it is difficult to simultaneously control the formation of heterocyclic ring and the atroposelective installation of a stereogenic axis[27–29]. In particular, the stereoselective construction of axially chiral isothiazole

S-oxides is more difficult because an extra stereogenic sulfur center affects the synthesis of structures bearing different types of chiralities[30,31]. Due to the structural versatility of atropisomeric 1,2-azoles and their potential functional diversity, it is highly desirable to design a platform molecule to access different types of axially chiral 1,2-azoles via novel heterocyclic ring formation.

In recent years, vinylidene *ortho*-quinone methide (VQM) intermediates have evolved into an important tool for catalytic asymmetric synthesis because their intrinsic axial chirality can be generated from readily available *ortho*-alkynylnaphthols with a chiral catalyst and further subjected to downstream transformations with excellent stereocontrol[32–38]. Based on our previous work on VQM chemistry for heterobiaryl atropisomer synthesis[39–43], we believed that VQMs could be an ideal template synthon for the collective synthesis of axially chiral 1,2-azoles. Nevertheless, reported VQMs are mostly limited to those bearing an arene ring at 5-position, so axial chiral architectures were

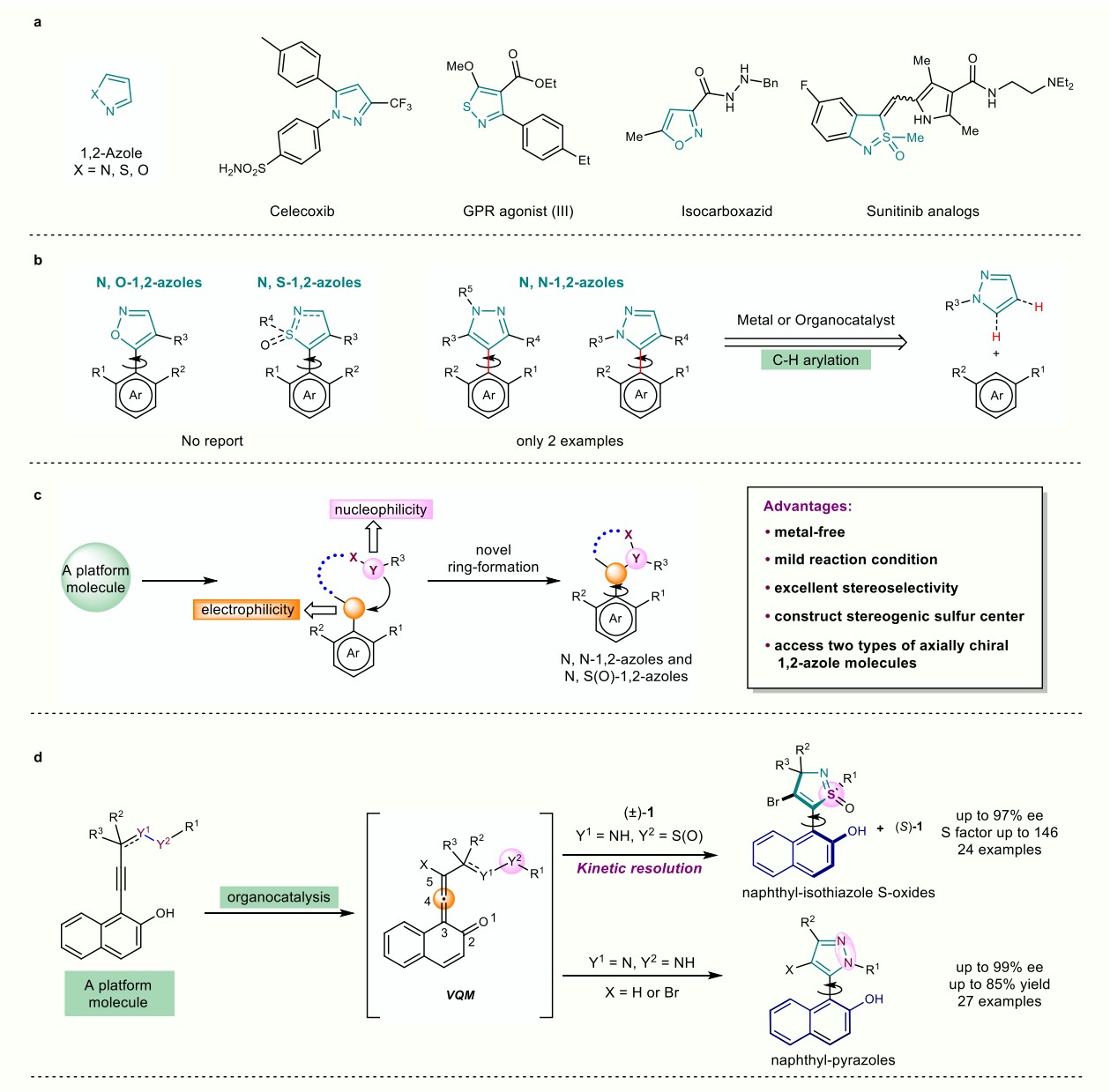

**Fig. 1 Background and project synopsis. a** Examples of 1,2-azole structures in drugs and bioactive molecules. **b** Synthesis status of axially chiral compounds containing 1,2-azole structures. **c** Our strategy: novel construction of axially chiral compounds containing 1,2-azoles. **d** This work.

confined to benzo- or naphtho-fused analogues[39–44]. To realize the synthesis of atropisomeric 1,2-azoles, we intended to design other types of VQM precursor bearing non-arene groups at 5-position. We envisioned that the open chain motif bearing two heteroatoms would be primed for subsequent cyclization and offer two extra sites for catalyst binding, thus forming an ideal entry point into the preparation of axial chiral 1,2-azoles (Fig. 1c).

Herein, we report our recent progress toward the construction of two types of axially chiral 1,2-azoles and their potential applications in synthetic transformations and therapeutic usage (Fig. 1d). First, racemic mixtures of N-naphthyl propargylsulfi-namides bearing a stereogenic sulfur center are used as substrates to undergo a kinetic resolution process with high S factors. Both products and substrates can be transformed into a set of atro-pospecific functional biaryls with potential applications as tem-plate molecules in asymmetric synthesis. Next, axially chiral naphthyl-pyrazoles are successfully constructed via modified VQMs bearing hydrazone moiety in high yields with good atro-poselectivities. Furthermore, we prove that some atropisomeric pyrazole agents possessed promising antiproliferation activity and relative safety, by in vitro tests on six human cancer cell lines and one normal cell line.

## Results

**Reaction development of naphtyl-isothiazole S-oxides.** Our attention first moved toward the enantioselective construction of isothiazole S-oxides, which are also known as cyclic sulfoximines and belong to an underutilized class of heterocycles. Their nucleophilic and basic nitrogen atom and stereogenic center at the sulfur atom make them a promising framework for biological and catalytic applications[45–50]. To date, the catalytic asymmetric preparation of stereogenic C-C axial chiral aryl-isothiazole S-oxides remains unexplored. Possible methods to address this challenge may involve the formation of axis and stereogenic sulfur center, but the simultaneous control of the stereogenic sulfur center and axis has not been achieved[30,51–59]. Our current strategy involves the VQMs bearing a racemic sulfinamide moi-ety, which can be deprotonated to form a carbon electrophile and then undertake an intramolecular S–C bond formation to access axially chiral naphtyl-isothiazole S-oxides through kinetic reso-lution, and recovering one isomer of sulfinamides 1 as stereogenic sulfur center derivatives.

**Condition screening.** First, we examined the reaction of (±)−1a and NBS in toluene with various organocatalysts at −40 °C for 0.5 h. Cinchona alkaloids-derived thiourea catalysts **A** and **B** provided poor conversion and enantioselectivity (Fig. 2c). Grat-ifyingly, cinchona alkaloids-derived squaramides (**C**, **D**, **E** and **F**) led to more enantiomerically pure products and afforded satis-factory results in the formation of isothiazole S-oxide **2a** (up to 42% conversion, 83% ee, S value up to 20). By contrast, the amide catalysts **G** and **H** resulted in the lower catalytic per-formance. With the best catalyst **F** in hand, solvents including m-xylene, iPrOH, acetonitrile (CH3CN), ethyl acetate (EA), tet-rahydrofuran (THF) and dichloromethane (DCM) were investi-gated, but none of them outperformed DCM in performance (S = 25). The further evaluation of reaction temperature sug-gested that the highest S value could be obtained under the reaction conditions as follow: 0.05 mmol substrates **1**, 0.0275 mmol NBP and 10 mol% catalyst **F** in DCM (2.0 mL) at −78 °C for 0.5 h.

**Substrates scope exploration of naphtyl-isothiazole S-oxides.** With the optimized reaction conditions in hand, we investigated the substrate scope of the established kinetic resolution of N-propargylsulfinamides to form axially chiral naphtyl-isothiazole S-oxides through enantioselective annulation of VQM inter-mediates (Fig. 3a). The substrates with tert-butylsulfinamides motif at 5-position of VQM precursor allowed the kinetic reso-lution process to give chiral heterocycles **2a–2h** in high S values with up to 96% ee and the corresponding sulfinamides **1a–1h** were also obtained with moderate to good enantioselectivities. Mean-while, replacing the tert-butylsulfinamide with less hindered iso-propyl sulfinamide at 5-position led to lower performance in terms of enantioselectivity and conversion (**2j**, S = 4, ee = 52%). Furthermore, benzosulfinamides substituted substrates were well tolerated to the reaction conditions, thus affording chiral naphthyl-isothiazole S-oxides (**2k–2t**) with up to 97% ee. How-ever, the electron-withdrawing group on the phenyl part of sul-finamides led to the decreased performance (**2n**, S = 4, ee = 55%; **2o**, S = 1, ee = 11%). It is worth noting that both electron-donating and electron-withdrawing substituents can be installed to the naphtyl ring of the substrates **1** without significant changes in enantioselectivities and conversions. Substrates with the naphtyl group replaced by quinoline also proceeded smoothly under standard reaction conditions, giving the quinolinyl-isothiazole S-oxides (**2i** and **2s**) and sulfinamides **1i** and **1s** in an enantiopure form. However, substrates **1** with other gem-disubstituted groups [e.g, dihydro (**2u**), diethyl (**2w**), cyclohexyl (**2x**)] attached to the propargylic carbon provides the target isothiazole derivatives with extremely low yields and poor enantioselectivities under the standard reaction conditions (See Supplementary Information for details). The absolute configurations of **1e** and **2e** were determined to be (S) and (aS, S) by X-ray crystal structure analysis and others were assigned by analogy.

**Synthetic applications.** Next, the synthetic potential of our reaction was then explored (Fig. 3b). A gram-scale reaction was firstly carried out to prepare **1k** and **2k** under the optimal reac-tion conditions and the yield and enantioselectivity almost showed no change. The ee value of substrate **1k** and the atropi-someric **2k** could be further improved to 99% after recrystalli-zation. In addition, chiral sulfinamide **1k** and axially chiral naphthyl-isothiazole S-oxide **2k** were found diverse transforma-tions. **1k** could be converted to corresponding naphthyl-isothiazole S-oxides **1ka–1kd** in the presence of NIS, NBP or phenylthiohypochlorite and phenylselenenylchloride at corre-sponding temperature in DCM with good yields and enantios-electivities. The hydroxyl group of **2k** could be esterified to give products **2ka**, **2kb** and **2kc** in excellent yields without erosion in optical purity.

**Reaction development of naphthyl-pyrazoles.** After the estab-lishment of protocol toward atroposelective construction of naphthyl-isothiazole S-oxides, we turned our attention to explore the protocol toward the preparation of axial chiral pyrazole derivatives based on VQMs bearing non-arene groups at 5-position. Pyrazole is a common privileged substructure in both natural and artificial functional molecules[60–66]. To our surprise, catalytic synthetic approach has not been developed to construct the stereogenic axis and pyrazole ring simultaneously. In this study, we conceived to design another VQM precursor bearing hydrazone-substituted substrate applicable for the axial chiral pyrazole formation (Fig. 4). Primary screening results suggested that sulfonylbenzenes or diphenylphosphine oxides were applic-able to the reaction to provide the steric hindrance required for the formation of atropisomers. After screening reaction con-ditions, optimal reaction conditions for sulfonyl hydrazones substrates were identified as follows: 0.05 mmol substrates and

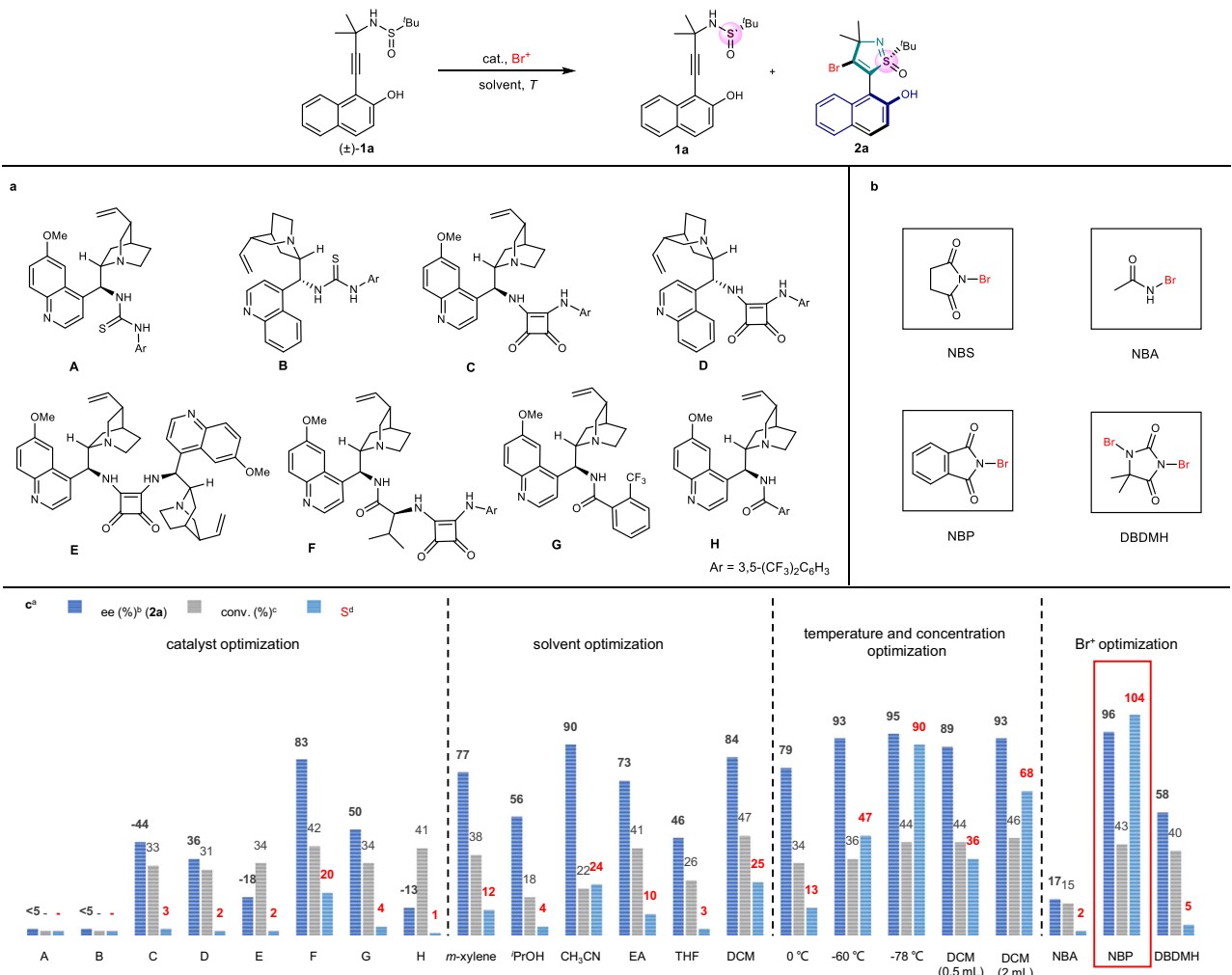

**Fig. 2 Optimization of the reaction conditions. a** Catalysts. **b** Brominating reagents: *N*-bromosuccinimide (NBS); *N*-bromoacetamide (NBA); *N*-bromophthalimide (NBP); 1,3-dibromo-5,5-dimethylhydantoin (DBDMH). **c** Reaction conditions screening. ª(±)−**1a** (0.05 mmol), catalysts **A**–**H** (10 mol%) in solvent at corresponding temperature for 10 min, then brominating reagents (0.0275 mmol) were added at corresponding temperature, unless otherwise specified. ᵇEnantiomeric excess (ee) were determined by HPLC. ᶜConversion ratio was calculated by the methods of Fiaud: conv. = ee$_{1a}$/(ee$_{1a}$ + ee$_{2a}$). ᵈSelectivity values were calculated by the methods of Fiaud: S = ln[(1 – conv.)(1 – ee$_{1a}$)]/ln[(1 – conv.)(1 + ee$_{1a}$)]. ᵗBu *tert*-Butyl.

10 mol% catalyst **C** in DCM (2.0 mL) at 25 °C for 12 h (For details, see Supplementary Table 2).

**Substrates scope exploration of naphthyl-pyrazoles**. To further test the substrate scope, various enantioselectivities and good yields were observed in most cases (Fig. 4a). For the preparation of axially chiral naphthyl-pyrazoles, alkyl substitutions including *tert*-butyl (**4a**), methyl (**4b**), ethyl (**4c**), isopropyl (**4d**), 3-pentyl (**4e**), cyclopentyl (**4f**), cyclohexyl (**4g**), and more steric 1-adamantyl (**4h**), as well as phenyl (**4i**) on the 3-position of pyrazole, were afforded in high yields and enantioselectivities (72–85% yields, 83–99% ee). For the sulfonylbenzene ring, non-substitution (**4j**) or the substitution of methyl (**4a**), trifluoromethyl (**4m**), or chloro (**4n**) on the 4-position caused no significant variation in both yields and ee. Particularly, the substrates with alkyl groups gave rise to slightly higher enantiomeric purity on average, as indicated by the comparison of **4a**-**4h** (>88% ee), **4i** (83% ee), **4k** (84% ee) and **4l** (94% ee). Besides, brominating the 7-position of the 2-naphthol ring did not obviously affect the yields or enantiopurities (**4o** and **4p**). In addition, the absolute configuration of **4a** was detected to be (*aR*) by X-ray crystallographic analysis and others were assigned by analogy.

Similar optimizations were also carried out for the synthesis of the other axially chiral pyrazoles bearing the diphenylphosphine oxide moiety and satisfying results were pleasingly acquired within a short reaction time (Supplementary Table 3). For this type of axial chiral naphthyl-pyrazole, a wide substrate scope was also observed (Fig. 4b). *tert*-Butyl (**4q**), 3-pentyl (**4r**), cyclopropanyl (**4s**), cyclobutyl (**4t**), cyclohexyl (**4u**), steric 1-adamantyl (**4v**), and *p*-tolyl (**4w**) substituted *N*-diphenylphosphine oxide pyrazoles were acquired in moderate yields with good to excellent ee (70–78% yields, 86–98% ee). We further investigated the substrates with Br and OMe substituted in naphthyl moiety, and the reactions proceeded quite smoothly (**4x**-**4aa**, 70–75% yields, and 85–97% ee). In addition, the absolute configuration of **4q** was detected to be (*aS*) by single-crystal X-ray crystallographic analysis and others were assigned by analogy.

**Biological activity study**. Another goal of our research is to explore latent medicinal utilization for the chiral naphthyl-pyrazoles. A physicochemical analysis on the synthetic method suggested that many of the naphthyl-pyrazoles followed Linpiski's "rule of 5" basically and therefore possessed drug-like properties

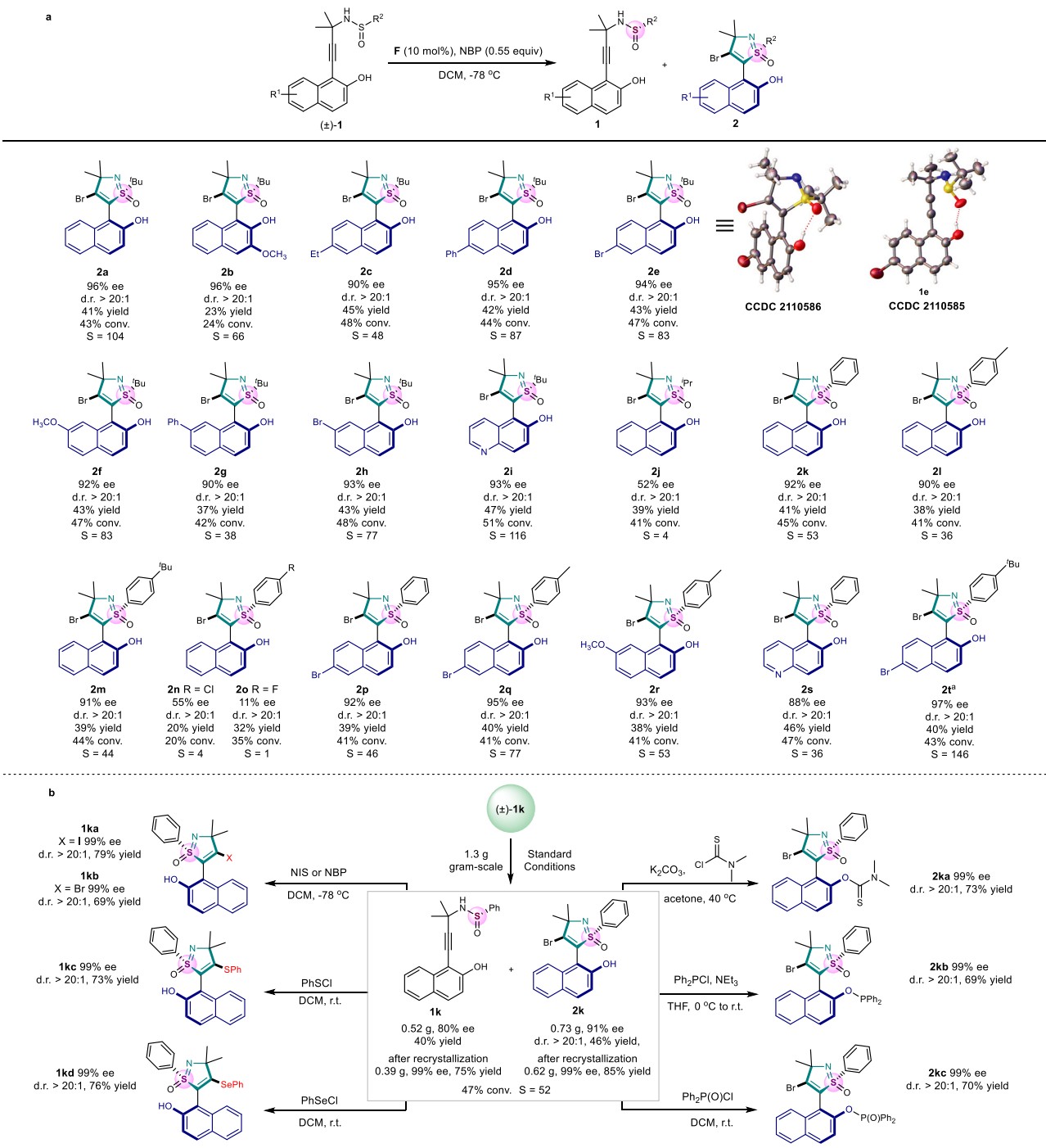

**Fig. 3 Substrate scope of naphthyl-isothiazole S-oxides and synthetic transformations. a** Substrate scope of naphthyl-isothiazole S-oxides. Reaction conditions: (±)−**1** (0.3 mmol), **F** (10 mol%) in DCM (12.0 mL) at −78 °C for 10 min, then NBP (0.165 mmol) were added; Enantiomeric excess (ee) were determined by HPLC; Diastereomeric ratio (d.r.) was determined by HPLC; Isolated yield; Conversion ratio was calculated by the methods of Fiaud: conv. = ee$_1$/(ee$_1$ + ee$_2$). Selectivity values were calculated by the methods of Fiaud: S = ln[(1 − conv.)(1 − ee$_1$)]/ln[(1 − conv.)(1 + ee$_1$)], unless otherwise specified. [a] **F** (20 mol%). **b** Gram-scale reaction and synthetic transformations (NIS: N-Iodosuccinimide). [t]Bu tert-Butyl, [i]Pr isopropyl.

(see Supplementary Discussion). Subsequently, a set of biological assays were performed to evaluate the potential of the axially chiral naphthyl-pyrazoles in the pharmacological application. Though some of the products resembled celecoxib in structure, they exhibited no inhibition on the COX-2 activity at concentrations up to 100 $\mu$M (**4m** and **4k**). Instead, many compounds exhibited obvious antiproliferation effects toward a small panel of cancer cell lines including A549, MIA PaCa-2, A375, MCF7, MDA-MB-231 and Hela. One of the optimal compounds,

**4x**, inhibited A375 cell by IC$_{50}$ of 2.57 $\mu$M (Fig. 5a) and its cytotoxicity toward non-cancer cell line L02 was only modest (25.01 $\mu$M). An interesting "activity cliff" was observed as the methoxyl group on C7-position of naphthoquinone obviously increased the activity (non-substituted **4s** with IC$_{50}$ of 28.15 $\mu$M, Fig. 5b). Meanwhile, we found that the two enantiomers were similar in bioactivity, where **4x** was slightly more potent than its enantiomer counterpart (IC$_{50}$ of 2.57 $\mu$M versus 3.6 $\mu$M, Supplementary Table 7). These observations provided theoretical

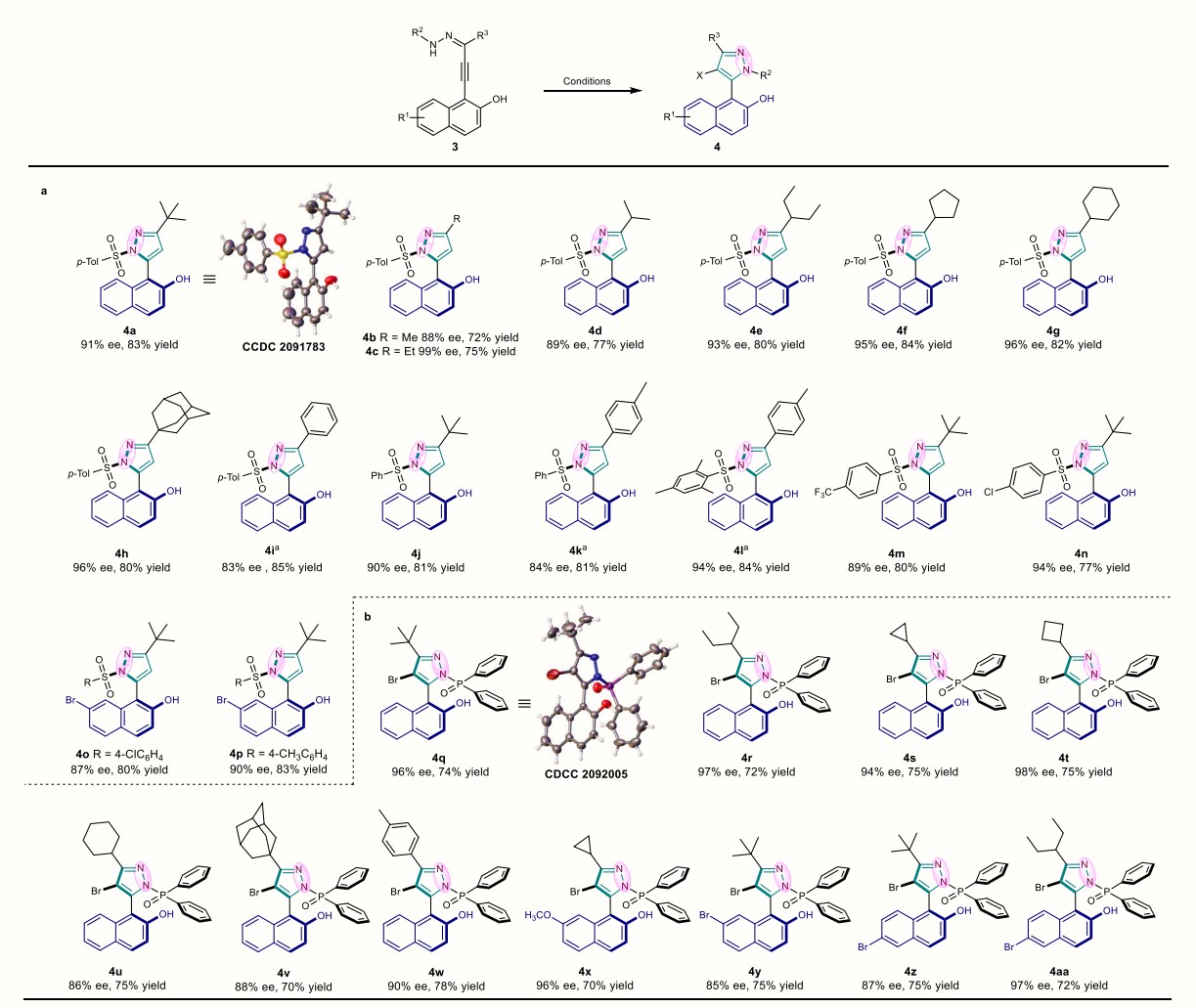

**Fig. 4 Substrate scope of naphthyl-pyrazoles. a** Reaction conditions: **3a–3p** (0.1 mmol), **C** (10 mol %) in DCM (4.0 mL) at 25 °C for 12 h; [a]Reaction time was extended to 24 h. **b** Reaction conditions: **3q–3aa** (0.1 mmol), **E** (20 mol %), NBS (0.105 mmol) in toluene (2.0 mL) at 25 °C for 1 h; Enantiomeric excess (ee) were determined by HPLC; Isolated yield.

supports to our assumption that the chiral naphthyl-pyrazoles deserved further exploration. To further explore the pharmacological mechanism, systematic flow cytometric assays were performed with compound **4x** (Fig. 5c). The percentage of A375 cells in the $G_1$ period decreased from 61.5 to 49.4%, when cells were treated with gradient doses (0, 2, 4, and 8 $\mu$M) of **4x** for 24 h. Under a single dose (4 $\mu$M) of **4x**, the percentage of cells in the $G_1$ period dropped from 68.5 to 46.6% after different reaction time (0, 12, 24, and 48 h, Fig. 5c). Likewise, the percentage of total apoptotic A375 cells increased from 3.5 to 46.4% when treated with 8 $\mu$M of **4x** for 24 h, or increased from 8.0 to 41.1% when treated with 4 $\mu$M of **4x** for 48 h (Fig. 6a). It could be concluded from the results that compound **4x** caused cell cycle arrest and induced cell apoptosis in A375 cells in both time- and dose-dependent manners. As **4x** provoked a significant apoptosis effect, we explored whether it was ascribed to the production of ROS (reactive oxygen species), a causal factor of mitochondrial-dependent apoptosis. The 48 h exposure of A375 cells to **4x** in different doses (2, 4, and 8 $\mu$M) led to an increased ROS generation in a dose-dependent manner, compared to the non-treated group (Fig. 6b). We also investigated the mitochondrial membrane potential (MMP) loss, an early hallmark of mitochondrial dysfunction, in A375 cells with the treatment of **4x**. The

mitochondrial depolarization percentage was found to rise from 7.6% of the non-treated control to 21.9% (2 $\mu$M), 59.8% (4 $\mu$M), and 77.4% (8 $\mu$M), respectively (Fig. 6c). Afterward, the mitochondrial apoptotic mechanism was further validated by the western blot assays. The treatment of A375 cell with **4x** for 48 h obviously elevated the expression levels of cleaved caspase-3, cleaved caspase-9, cleaved PARP, Bax and cytochrome c, and gradually decreased the expression levels of Bcl-2 and Bcl-xL in a dose-dependent manner (Fig. 6d). In summary, we proved that compound **4x** was an effective antiproliferation agent and induced apoptosis in A375 cells through the mitochondria-related pathways. Therefore, it may be developed for therapeutic purposes and further optimizing research on **4x** is in progress.

**Mechanistic investigations.** In order to demonstrate the formation of this type of novel VQMs and their crucial roles in atroposelective transformation, some control experiments were carried out. When the naphthyl hydroxyl was protected by acetyl group (**1l'**, **3i'**, **3q'**) or replaced with hydroxymethyl group (**1l"**) to prevent the formation of VQM intermediate, the reactions failed to deliver the desired heterobiaryl architectures under optimized reaction conditions even with prolonged reaction time

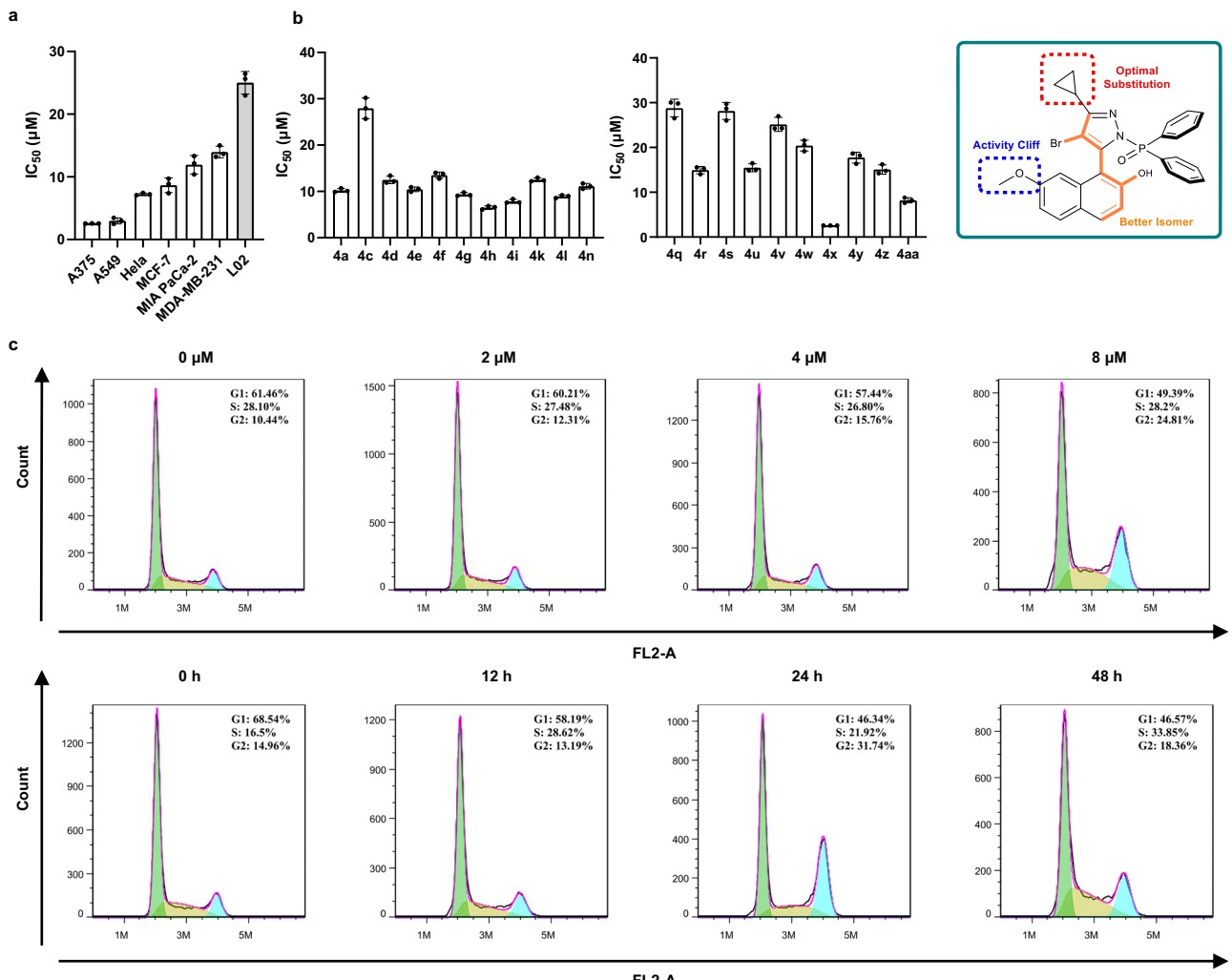

**Fig. 5 Pharmacological research on the axially chiral naphthyl-pyrazoles. a** The IC$_{50}$ values of **4x** toward different cell lines. **b** The IC$_{50}$ values of tested azoles toward A375 cells and structure-activity findings. **c** Compound **4x** induced A375 cell cycle arrest both in the dose- and time-dependent manners.

(Fig. 7a). The failures indicated that the formation of VQM intermediate was vital for this transformation and that VQMs were likely involved in the reaction process. Based on the above-mentioned control experiments, a plausible mechanism was depicted in Supplementary Fig. 6, the high stereoselectivity of the annulation could be justified with a model reported by our group previously[42]. Moreover, taking the axially chiral naphthyl-isothiazole S-oxides (N, S-1,2-azoles) as an example, the favored transition states were proposed in Fig. 7b. The ring formation step possibly favors the intramolecular nucleophilic attack of the VQM intermediates from less sterically hindered side, and subsequent cyclization affords **2** as the target products.

## Discussion

In conclusion, we developed two types of VQM precursor containing the substituents of two heteroatoms (sulfinamides and hydrazones) at 5-position and obtained axially chiral naphthyl-isothiazole S-oxides (N, S-1,2-azoles) and naphthyl-pyrazoles (N, N-1,2-azoles) in good to excellent yields and with high enantioselectivities via an organocatalytic novel ring formation process. First, racemic mixtures of *N*-naphthyl propargylsulfina-mides bearing a stereogenic sulfur center were used as substrates to undergo a kinetic resolution process with high S factors. Both products and substrates could be transformed into a set of atropospecific functional biaryls with potential applications as

template molecules in asymmetric syntheses. Next, axially chiral naphthyl-pyrazoles have been successfully constructed via VQMs bearing hydrazone moiety in high yields with good atroposelec-tivities. Furthermore, we proved that some atropisomeric pyr-azole agents possessed promising antiproliferation activity and relative safety, by in vitro tests on six human cancer cell lines and one normal cell line. Further applications of these types of VQM in atroposelective transformations will be reported in due course.

## Methods

**Synthesis of 2a–2x by asymmetric intramolecular annulation.** (±)−**1** (0.3 mmol, 1.0 equiv), and catalyst **F** (0.01 mmol, 10 mol%), DCM (12.0 mL) was injected into a flame-dried Schlenk tube; after stirring for 10 min at −78 °C, the NBP (0.165 mmol, 0.55 equiv) was added into the mixture. After completion of the reaction (monitored by thin-layer chromatography, TLC), the reaction was quenched with saturated ammonium chloride solution, and extracted with DCM (3 × 5.0 mL), all organic layer was removed under reduced pressure, and the residue was purified by column chromatography (PE: EA = 3:1–6:1) to afford product **2a–2x**.

**Synthesis of 4a–4p by asymmetric intramolecular annulation.** A flame-dried Schlenk tube equipped with a magnetic stirring bar, was charged with **3a–3p** (0.1 mmol, 1.0 equiv), and catalyst **C** (0.01 mmol, 10 mol%), DCM (4.0 mL) was injected into the tube at 25 °C. After stirring for 12 h, the reaction was quenched with saturated ammonium chloride solution, and extracted with DCM (3 × 4.0 mL), all organic layer was removed under reduced pressure, and the residue was purified by column chromatography (PE: EA = 4:1) to afford product **4a–4p**.

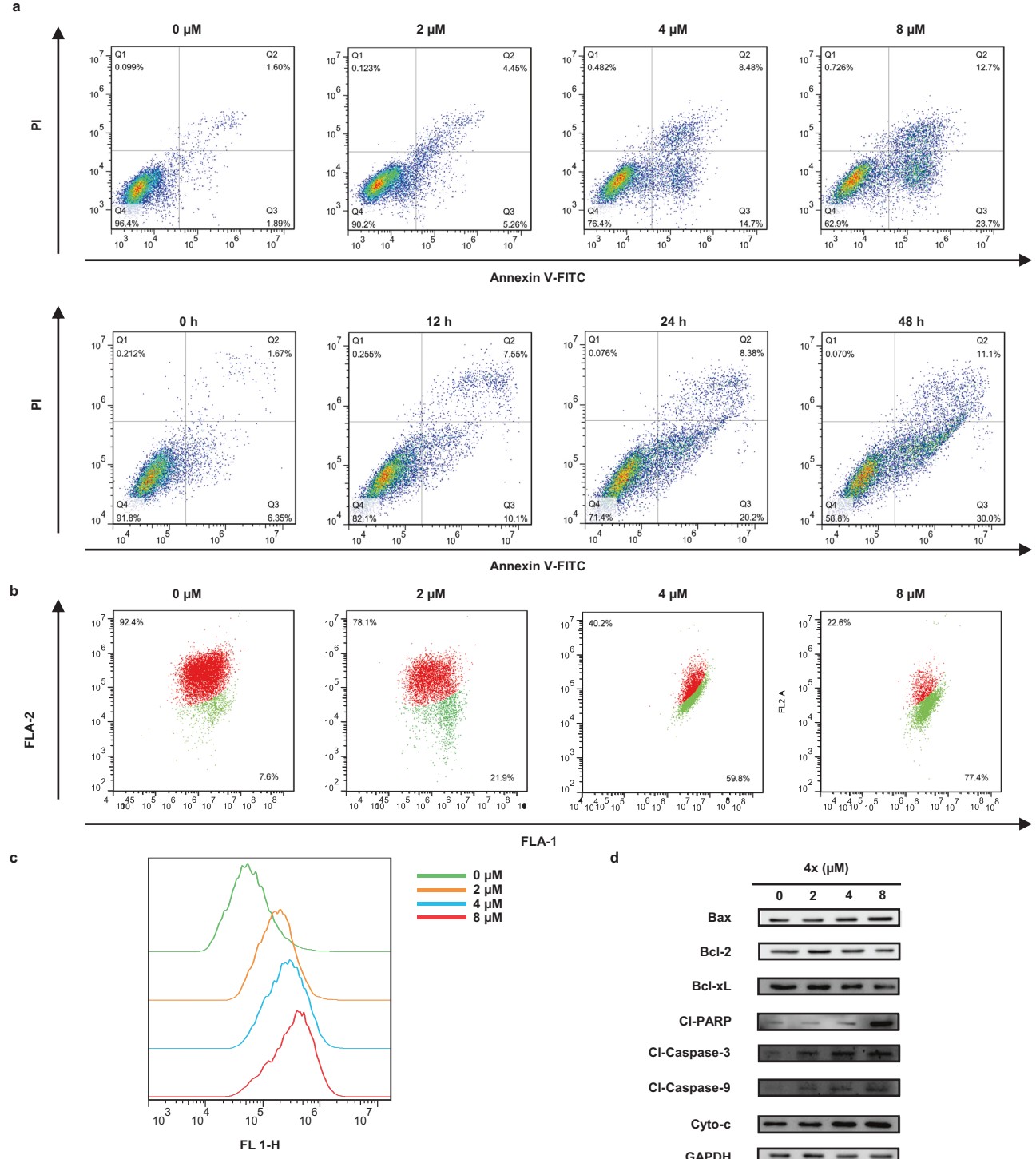

**Fig. 6 Compound 4x induced apoptosis in A375 cells through the mitochondria-related pathways. a** Compound **4x** induced A375 cell apoptosis both in the dose- and time-dependent manners. **b** Compound **4x** caused MMP loss in A375 cells in a dose-dependent manner. **c** Compound **4x** elevated the ROS level in A375 cells in a dose-dependent manner. **d** The expression levels of apoptosis-related proteins were assayed by western blot analysis. Data are presented as the mean ± SD from three independent experiments.

**Synthesis of 4q–4aa by asymmetric intramolecular annulation**. A flame-dried Schlenk tube equipped with a magnetic stirring bar, was charged with **3q–3aa** (0.1 mmol, 1.0 equiv), NBS (0.105 mmol, 1.05 equiv), and catalyst **E** (20 mol%), toluene (2.0 mL) was injected into the tube at 25 °C. After stirring for 1 h, the reaction was quenched with saturated ammonium chloride solution, and extracted with EA (3 × 2.0 mL), all organic layer was removed under reduced pressure, and the residue was purified by column chromatography (PE: EA = 3:1) to afford product **4q–4aa**.

## Data availability

Data relating to the characterization data of materials and products, general methods, optimization studies, mechanistic studies, mass spectrometry, HPLC and NMR spectra, computational studies, and biological activity studies are available in the Supplementary Information. Crystallographic parameters for compounds **1e**, **2e**, **4a** and **4q** are available free of charge from the Cambridge Crystallographic Data Centre under CCDC 2110585 (**1e**), CCDC 2110586 (**2e**), CCDC 2091783 (**4a**) and CCDC 2092005 (**4q**). These data can be obtained free of charge from The Cambridge Crystallographic Data Centre. All other

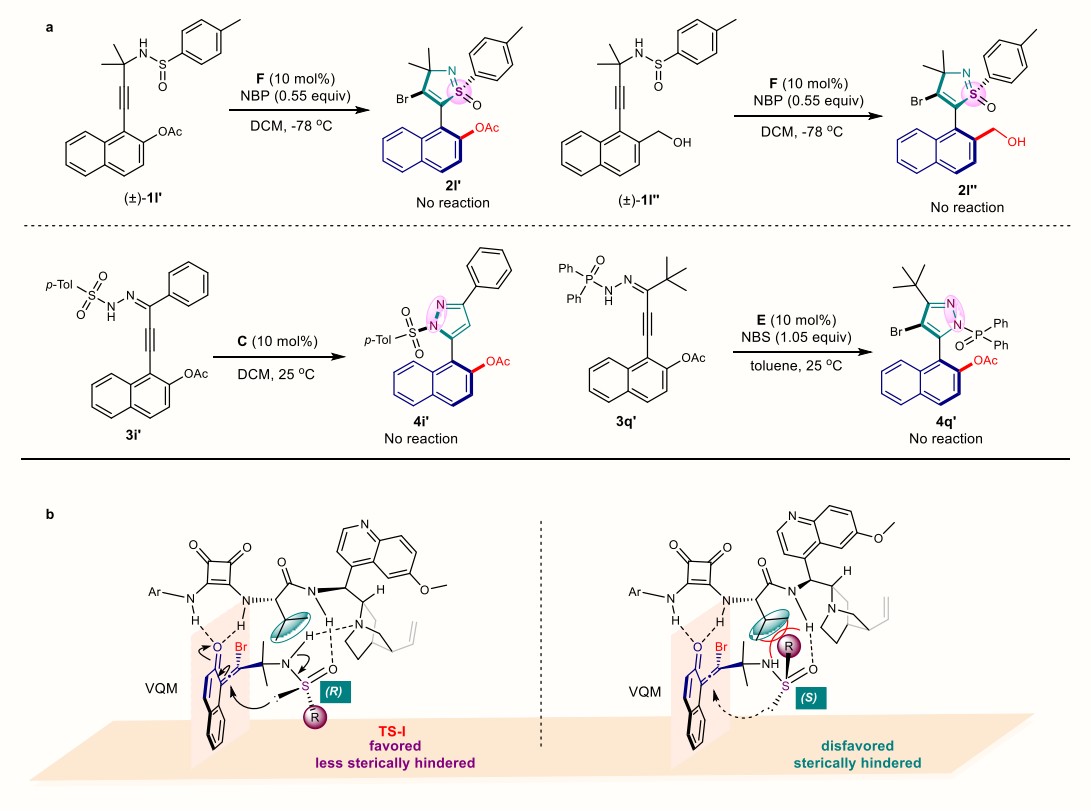

**Fig. 7 Control experiments and proposed mechanism model. a** Control experiments. **b** Proposed model for atroposelective 1,2-azole ring formation (naphthyl-isothiazole S-oxides as example).

data are available from the corresponding author upon request. Source data are provided with this paper.

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

## Acknowledgements

This study was supported by National Natural Science Foundation of China (Grant No: 21922101, 21772018, 21901026, 22001025) and the Fundamental Research Funds for the Central Universities (Project No. 2020CDJQY-Z002, 2021CDJQY-035). We thank Mr. Xiangnan Gong (Analytical and Testing Center of Chongqing University) for X-ray crystallographic analysis.

## Author contributions

H.Y. and W.Q. conceived and directed the project. Y.C. and C.X. designed and performed experiments and prepared the Supplementary Information. C.X. and P.W. performed the bioactive investigations. Y.C., C.X., H.L., S.H., and P.W. analyzed and interpreted the experimental data. Y.C. and C.X. contributed equally to this project. H.Y., W.Q., and P.W. wrote the paper. All authors discussed the results and commented on the paper.

## Competing interests

The authors declare no competing interests.
