## [Peer Review File · Nature Communications]

REVIEWER COMMENTS

Reviewer #1 (Remarks to the Author):

The manuscript by Yan and Qin describes the enantioselective organocatalyzed synthesis of two families of five-membered atropisomers : axially chiral isothiazoles and pyrrazoles. The reaction exploits the reactivity of vinylidene quinone methide (VQM) intermediates, which are generated thanks to an organocatalyzed 1,5-proton shift, a well-developed strategy in their group. The originality of this work lies in the fact that a new type of VQM has been designed, in which the nucleophilic moiety is not carried by an aryl moiety, allowing the access to N,N- and N,S-1,2-azoles. In the case of the synthesis of S-1,2-azoles, an additional feature of the present methodology is the simultaneous control of the stereogenicity of the sulfur atom, thanks to a kinetic resolution step, thus affording the final heterocycle with two stereogenic elements. The access to pyrazole atropisomers is also interesting since only two examples of pyrazole atropisomers synthesis have been reported before, both using a complementary C-H functionalization strategy. The strategy is certainly of interest and I would support publication in Nature Communications if the following questions could be addressed in a revised manuscript.

- 1) the term "chiral axis" or "chiral sulfur atom" is often used in the manuscript, which is not correct. It should be replaced by "stereogenic axis" and "stereogenic sulfur atom".
- 2) page 3, first line "idea" should be replaced by "ideal"
- 3) a gem-dimethyl group is apparently necessary in the case of isothiazole, which severely limits the scope. Could the authors comment about this ? Would it be possible to switch to other gem-disubstituted groups ?
- 4) OH moiety should be redrawn for product 2s in Fig. 3
- 5) the enantiomerization barriers have not been determined for these new families of compounds. In particular compounds 4a-p should display rather low rotational barrier. I think it would give interesting data to determine the enantiomerization barrier of one member of each family of compound.
- 6) Have the authors envisaged to use the corresponding oxime substrates to access isoxazoles ? The enantiomerization barrier in this case may be too low, but interesting to compare it with pyrrazole and thiazoles, to see the effect of the heteroatom on the barrier value.
- 7) The antiproliferative activity of some synthesized compounds has been evaluated, and judged as "considerable" for one of them by the authors. I am not an expert in this field, and I can hardly judge the impact of this biological study of this manuscript. Nevertheless, it seems to me that micromolar concentrations are relatively high, and these results have, at these stage, rather "promising" than "considerable" impact.

Reviewer #2 (Remarks to the Author):

Reviewer (Remarks to the Author):

The manuscript submitted by Yan and co-workers reported a de-novo ring formation strategy for atroposelective synthesis of two types of axially chiral 1, 2 azoles (naphthyl-isothiazole S-oxides and atropoisomeric naphthyl pyrazoles) through modified vinylidene ortho-quinone methide (VQM) intermediates. The strategy reported in this manuscript features fairly broad substrate generality, high efficiency and enantiocontrol, and the de-novo ring formation strategy was first time applied for the preparation of diverse axially chiral 1,2 azoles. This protocol should be considered as a significant achievement in the enantioselective construction of five membered nitrogen containing heterocyclic atropisomers with diverse skeletons. Moreover, the simultaneous control of the sulfur chiral center and the axis for naphthyl-isothiazole S-oxides, and the systematic bioassay for naphthyl pyrazoles should be highly prized as well. Furthermore, the utility of naphthyl-isothiazole S-oxides products was also demonstrated through various

transformations. At the end, the authors proved that some atropisomeric pyrazole agents possessed considerable antiproliferation activity and relative safety, by in vitro tests on six human cancer cell lines and one normal cell line. Overall, this manuscript is carefully written, all the important aspects were convincingly discussed in the manuscript. I believe that the present work is of significance to the field of catalytic atroposelective synthesis of biaryl atropisomers, will be of great interest to the broad readership of Nature communications. Consequently, I will give my support for the acceptance of this manuscript to publish in Nature Communications after minor revision on the following issues:

Corrections/comments:

1. In the first paragraph on page 3, "Nevertheless, reported VQMs are mostly limited to those bearing an arene ring at 5-position, so axial chiral architectures were confined to benzo- or naphtho-fused analogues", the VQM intermediate in the Fig. 1 d should have systematic number so that we can know the 5 position.
2. In the Fig. 3 a on page 6, the last compound in the substrate scope of naphthyl-isothiazole S-oxides "2t[b]" should be "2ta", and "bF (20 mol%)" should be "aF (20 mol%)".
3. In the third paragraph on page 8, "see Supporting Information" should be "see Supplementary Information".
4. In supplementary information, page S3, in the step 1, "S2 (1.5 equiv) was dissolved in THF (3 mmol/mL)" should be "S2 (1.5 equiv) was dissolved in THF (3.0 mmol/mL)"; "NEt3 (5 equiv)" should be "NEt3 (5.0 equiv)".
5. In supplementary information, page S4, in the step 2, "This step was carried out according to a literature method[6],[7],[8]with" should be "This step was carried out according to some literature methods[6],[7],[8] with".
6. In supplementary information, page S9, in the step 1, "Dissolve crude S6 (1.0 equiv) and P,P-Diphenylphosphinic hydrazide in THF (1 mmol/mL)" should be "Dissolve crude S6 (1.0 equiv) and P,P-Diphenylphosphinic hydrazide in THF (1.0 mmol/mL)"; in the step 1, "The mixture stirred 15min in room temperature" should be "The mixture stirred 15 min in room temperature".
7. In supplementary information, page S56, HPLC analysis of compound 4x, the retention time of chiral compound 4x is inconsistent with racemic 4x.
8. The ¹H NMR spectra and ¹³C NMR spectra of compound 3f were missing.

Reviewer #3 (Remarks to the Author):

The content presented in the manuscript is from a well-done research. It presents a tandem research from the authors about their exciting recent articles on the high stereoselectivity outcome involving chiral VQM intermediates. In particular, they report here a novel synthetic approach to specific atropisomers in a very high enantioselective fashion. They went further and identified one of its novel compounds with good biological activity, in addition to considering its physicochemical properties. Hopefully, in the near future, they can identify a synthetically VQM-based drug in advanced preclinical stages, which demonstrate good in vivo efficacy, reasonable in vivo PK, and good safety profile.

Following are a few comments on the manuscript.

1. Chiral transformations from (S)-1k (Fig. 3b) need more rigorous stereochemical assignments. The chiral atropisomeric determinations would not be assigned as analogy from the transformations using organocatalysts (like others throughout the manuscript), but as a result of the intramolecular VQM chiral induction by the stereogenic sulfinamide group. Ideally, stereochemical assignment of any compound from the list of 1ka to 1kd would be obtained from X-ray crystallography. However, physical property comparison of anyone of these enantiomers with their corresponding counterparts could be enough. For example, although putative enantiomers 2k and 1kb are shown to have identical nmr spectra in SI (including traces of impurities), additional physical data would be desirable, such as demonstration that regular HPLC (not chiral HPLC) coelution of a made mixture of 1kb [obtained from (S)-1k] and (R)-2k would

- result in one single peak as evidence that they are not diastereomers. Also, in a different determination using chiral HPLC, spiking the corresponding racemic mixture with 1kb would show increase of one of the two peaks (rather than showing three peaks).
2. All sufinamide examples have a gem-dimethyl group at the propargylic carbon, have you done the reaction when the propargylic carbon does not have substituents (i.e, a CH₂), or when it is attached to only one methyl group, or is part of a ring, e.g., cyclopropyl, cyclopentyl, etc.?
 3. For the gram scale preparation of (aS, S)-2k and (S)-1k, please include the recrystallization yields for both compounds.
 4. Please either specify or not the stereogenic assignment when writing the ID of chiral compounds throughout the manuscript. For example, always when referring to compound 1k, write either (S)-1k or just 1k.
 5. Page 8, bottom. It is indicated that 4s IC₅₀ is 28.15 uM, however, I think, Fig. 5b shows 4s IC₅₀ is under 20 uM. Could you please correct it?
 6. Page 8, bottom. It is common that IC₅₀ potency/activity of enantiomers is different, so I would suggest replacing the statement "the two enantiomers were not exactly equivalent in bioactivity, for 4x was slightly more potent than its enantiomer counterpart (IC₅₀ of 2.57 μM versus 3.6 μM)" with "the two enantiomers were similar in bioactivity, where 4x was slightly more potent than its enantiomer counterpart (IC₅₀ of 2.57 μM versus 3.6 μM, Table S4)".
 7. Page 9. I think the phrase "performed with compound 4x (Fig. 5b)" should be replaced with "performed with compound 4x (Fig. 5c)".
 8. Page 11. I think the phrase "compared to the non-treated group (Fig. 6c)" should be replaced with "compared to the non-treated group (Fig. 6b)".
 9. Page 11. I think the phrase "77.4% (8 μM), respectively (Fig. 6d)" should be replaced with "77.4% (8 μM), respectively (Fig. 6c)".
 10. Page 11, top. Please replace "or replaced with methanol group" with "or replaced with hydroxymethyl group".
 11. Page 12, upper paragraph. Please replace "Moreover, take the axially chiral" with "Moreover, taking the axially chiral".
 12. Please consider replacing "de novo" with "novel" throughout the paper. To me, "novel" sounds more accurate.
 13. Please replace "1, 2 azoles" with "1,2-azoles".
 14. Page 8. I would suggest to replace "with alkyl groups gave rise to higher enantiomeric purity" with "with alkyl groups gave rise to slightly higher enantiomeric purity on average".
 15. Page 8. Please replace "were assigned as analogues" with "were assigned by analogy". This suggested correction occurs twice on the same page 8.

Point-by-point responses to referees for manuscript (NCOMMS-21-41274)

Reviewer #1

The manuscript by Yan and Qin describes the enantioselective organocatalyzed synthesis of two families of five-membered atropisomers: axially chiral isothiazoles and pyrrazoles. The reaction exploits the reactivity of vinylidene quinone methide (VQM) intermediates, which are generated thanks to an organocatalyzed 1,5-proton shift, a well-developed strategy in their group. The originality of this work lies in the fact that a new type of VQM has been designed, in which the nucleophilic moiety is not carried by an aryl moiety, allowing the access to N, N- and N, S-1,2-azoles. In the case of the synthesis of S-1,2-azoles, an additional feature of the present methodology is the simultaneous control of the stereogenicity of the sulfur atom, thanks to a kinetic resolution step, thus affording the final heterocycle with two stereogenic elements. The access to pyrazole atropisomers is also interesting since only two examples of pyrazole atropisomers synthesis have been reported before, both

using a complementary C-H functionalization strategy. The strategy is certainly of interest and I would support publication in Nature Communications if the following questions could be addressed in a revised manuscript.

We greatly appreciate the reviewer's comments on our work.

Comment 1:

The term "chiral axis" or "chiral sulfur atom" is often used in the manuscript, which is not correct. It should be replaced by "stereogenic axis" and "stereogenic sulfur atom".

Response:

We have changed the "chiral axis" and "chiral sulfur atom" into "stereogenic axis" and "stereogenic sulfur atom" respectively in the revised manuscript and highlighted them in yellow.

Comment 2:

Page 3, first line "idea" should be replaced by "ideal".

Response:

We have revised the "idea" into "ideal" in Page 3, first line.

Comment 2:

A gem-dimethyl group is apparently necessary in the case of isothiazole, which severely limits the scope. Could the authors comment about this? Would it be possible to switch to other gem-disubstituted groups?

Response:

We have examined a variety of gem-disubstituted substrates including dihydrogen **1u**, diethyl **1w** and cyclohexanyl **1x** in this transformation (Figure R1). Unfortunately, all of these gem-disubstituted substrates provides the target isothiazols with relatively low yields and poor enantioselectivities at the current stage. This information has been added at the paragraph 1 on page 5 of the revised manuscript.

Fig R1. The investigation of gem-disubstituted substrates

Comment 4:

OH moiety should be redrawn for product **2s** in Fig. 3

Response:

The OH moiety of product **2s** in Fig. 3 has been redrawn in the revised manuscript.

Comment 5:

The enantiomerization barriers have not been determined for these new families of compounds. In particular compounds **4a-p** should display rather low rotational barrier. I think it would give interesting data to determine the enantiomerization barrier of one member of each family of compound.

Response:

We thank the reviewer for raising this point and we have carefully investigated the enantiomerization barriers by testing the half-time of one member of each family of compound. As shown in Fig R2, the rotational barrier of isothiazole S oxides were relatively high, and the half-time of representative compound **2k** was determined to be 3465 hours. Instead, as the reviewer mentioned that, compounds **4a-4p** displayed low rotational barrier, the representative compound **4f** showed short half-time (half-life: 14 min at 90 °C and 169 min at 55 °C). Moreover, the axially chiral pyrazoles bearing the diphenylphosphine oxide moiety were shown to be unstable when heating up to 55 °C. These results have been added into our revised supplementary information.

Fig R2. The enantiomerization barrier investigation

Racemization studies of 2k

Fig R3. The enantiomerization barrier investigation of 2k

Thermal Racemization of 2k: A solution of **2k** (17.5 mg, 97% ee) in toluene (3 mL) was heated at 90 °C. At intervals, small samples were taken and the solvent was removed by evaporation. The enantiomeric excess was determined by using HPLC (HPLC conditions: Chiralcel OD-H, Hexane/*i*-PrOH = 85:15, flow rate = 1.0 mL/min, wave length = 254 nm).

Time/h	% second eluted enantiomer (%t)	ln ((%t-50)/(%0-50))
0	97.20	0
0.166	97.18	-0.00042
0.25	97.16	-0.00085
0.42	97.14	-0.00127
0.75	97.14	-0.00127
1	97.12	-0.00170
1.916	97.11	-0.00191
3.916	97.10	-0.00212
5.916	97.05	-0.00318
10.416	96.98	-0.00467
16.416	96.95	-0.00531
21.42	96.91	-0.00616
25.42	96.85	-0.00744
29.42	96.81	-0.00830
33.42	96.80	-0.00851
37.42	96.74	-0.00979
41.42	96.69	-0.01086
45.42	96.67	-0.01138
49.42	96.64	-0.01194
54.42	96.58	-0.01322
58.42	96.55	-0.01387
62.42	96.52	-0.01451
69	96.48	-0.01537

73	96.46	-0.01580
77	96.42	-0.01666
81	96.34	-0.01839
85	96.30	-0.01925
93	96.27	-0.01990
97	96.24	-0.02055

Racemization studies of 4f

Fig R4. The enantiomerization barrier investigation of 4f

Thermal Racemization of 4f: A solution of 4f (20 mg, 95% ee) in toluene (3 mL) was heated at given temperature. At intervals, small samples were taken and the solvent was removed by evaporation. The enantiomeric excess was determined by using HPLC (HPLC conditions: Chiralcel OD-H, Hexane/*i*-PrOH = 90:10, flow rate = 1.0 mL/min, wave length = 254 nm).

T = 90 °C

Time/min	% second eluted enantiomer (%t)	$\ln ((\%t-50)/(\%0-50))$
0	95	0
2	92.52	-0.05802
7	86.02	-0.22392
12	79.5	-0.4236
17	73.74	-0.64083
22	63.95	-1.17252
27	58.18	-1.70630
32	51.68	-3.28920

T = 55 °C

Time/min	% second eluted enantiomer (%t)	$\ln ((\%t-50)/(\%0-50))$
0	95	0
10	94.1	-0.0202
20	92.35	-0.06069
30	91.1	-0.09065
40	90.4	-0.10783
50	88.55	-0.15471
60	87.85	-0.17303
70	86.34	-0.21374
80	84.62	-0.26223
90	83.98	-0.28089
100	82.82	-0.31562
110	81.48	-0.35731
120	80.3	-0.39551
130	78.9	-0.44282
140	77.56	-0.49030
150	76.48	-0.53027
160	74.84	-0.59421
170	73.94	-0.63111
185	72.56	-0.69048
200	71.1	-0.75739
220	69.08	-0.85802
240	67.38	-0.95134
270	64.76	-1.11474

Racemization studies of N-diphenylphosphine oxide pyrazoles 4x: The N-diphenylphosphine oxide pyrazole compound 4x were unstable and decomposes under heating conditions.

Comment 6:

Have the authors envisaged to use the corresponding oxime substrates to access isoxazoles ? The enantiomerization barrier in this case may be too low, but interesting to compare it with pyrazole and thiazoles, to see the effect of the heteroatom on the barrier value.

Response:

Thank you very much for this interesting suggestion, and we have prepared an oxime substrate **5** and subjected to standard reaction conditions for preparing racemic products, and we have successfully obtained corresponding isoxazole compound **6**, the product was identified through ¹H-NMR, ¹³C-NMR and MS. (Fig R5, identification data please see appendix part). However, the attempts to separate atropomers by chiral HPLC using different chiral columns were failed, indicating the enantiomerization barrier in this case may be too low as expected. To further understand the atropisomeric behaviour of this kind of hetro biaryls, we have performed a preliminary theoretical calculation to estimate the enantiomerization barrier, as a result, the estimated enantiomerization barrier was calculated to be 15.26 Kcal/mol using method B3LYP/6-31G(d) in Gaussian 09 program, indicating that such kind of isoxazole derivative was extremely easy to undergo the racemization (Fig R6).

Fig R5: The synthesis of isoxazole containing biaryl derivative

Fig R6: The preliminary evaluation of enantiomerization barrier of compound 6.

Comment 7:

The antiproliferative activity of some synthesized compounds has been evaluated, and judged as “considerable” for one of them by the authors. I am not an expert in this field, and I can hardly judge the impact of this biological study of this manuscript. Nevertheless, it seems to me that micromolar concentrations are relatively high, and these results have, at these stage, rather “promising” than “considerable” impact.

Response:

We thank the reviewer for raising this point and we have carefully rephrased the corresponding descriptions in our manuscript.

Reviewer #2

The manuscript submitted by Yan and co-workers reported a de-novo ring formation strategy for atroposelective synthesis of two types of axially chiral 1, 2 azoles (naphthyl-isothiazole S-oxides and atropisomeric naphthyl pyrazoles) through modified vinylidene ortho-quinone methide (VQM) intermediates. The strategy reported in this manuscript features fairly broad substrate generality, high efficiency and enantiocontrol, and the de-novo ring formation strategy was first time applied for the preparation of diverse axially chiral 1,2 azoles. This protocol should be considered as a significant achievement in the enantioselective construction of five membered nitrogen containing heterocyclic atropisomers with diverse skeletons. Moreover, the simultaneous control of the sulfur chiral center and the axis for naphthyl-isothiazole S-oxides, and the systematic bioassay for naphthyl pyrazoles should be highly prized as well. Furthermore, the utility of naphthyl-isothiazole S-oxides products was also demonstrated through various transformations. At the end, the authors proved that some atropisomeric pyrazole agents possessed considerable antiproliferation activity and relative safety, by in vitro tests on six human cancer cell lines and one normal cell line. Overall, this manuscript is carefully written, all the important aspects were convincingly discussed in the manuscript. I believe that the present work is of significance to the field of catalytic atroposelective synthesis of biaryl atropisomers, will be of great interest to the broad readership of Nature communications. Consequently, I will give my support for the acceptance of this manuscript to publish in Nature Communications after minor revision on the following issues:

We greatly appreciate the reviewer's comments on our work.

Comment 1:

In the first paragraph on page 3, "Nevertheless, reported VQMs are mostly limited to those bearing an arene ring at 5-position, so axial chiral architectures were confined to benzo- or naphtho-fused analogues", the VQM intermediate in the Fig. 1 d should have systematic number so that we can know the 5 position.

Response:

We thank the reviewer for pointing this out and we have added systematic number of VQM intermediate in Fig 1d.

Comment 2:

In the Fig. 3 a on page 6, the last compound in the substrate scope of naphthyl-isothiazole S-oxides "2t[b]" should be "2ta", and "bF (20 mol%)" should be "aF (20 mol%)".

Response:

We have corrected these typo errors in our manuscript.

Comment 3:

In the third paragraph on page 8, "see Supporting Information" should be "see Supplementary Information".

Response:

We have revised the "see Supporting Information" to "see Supplementary Information" in third paragraph on page 8 and we have proofed similar descriptions all over the manuscript.

Comment 4:

In supplementary information, page S3, in the step 1, “S2 (1.5 equiv) was dissolved in THF (3 mmol/mL)” should be “S2 (1.5 equiv) was dissolved in THF (3.0 mmol/mL)”; “NEt3 (5 equiv)” should be “NEt3 (5.0 equiv)”.

Response:

We have corrected these typo errors in our manuscript.

Comment 5:

In supplementary information, page S4, in the step 2, “This step was carried out according to a literature method[6],[7],[8]with” should be “This step was carried out according to some literature methods[6],[7],[8] with”.

Response:

We thank the reviewer and we have corrected these typos and grammar errors in our manuscript.

Comment 6:

In supplementary information, page S9, in the step 1, “Dissolve crude S6 (1.0 equiv) and P,P-Diphenylphosphinic hydrazide in THF (1 mmol/mL)” should be “Dissolve crude S6 (1.0 equiv) and P,P-Diphenylphosphinic hydrazide in THF (1.0 mmol/mL)”; in the step 1, “The mixture stirred 15min in room temperature” should be “The mixture stirred 15 min in room temperature”.

Response:

We have corrected these typos and grammar errors in our manuscript.

Comment 7:

In supplementary information, page S56, HPLC analysis of compound 4x, the retention time of chiral compound 4x is inconsistent with racemic 4x.

Response:

We thank the reviewer for pointing this out and we have re-examined the HPLC analysis of compound 4x, the corresponding HPLC spectrum have been added to the revised supplementary information part.

Comment 8:

The ^1H NMR spectra and ^{13}C NMR spectra of compound 3f were missing.

Response:

We thank the reviewer for pointing this out and we have added the ^1H -NMR spectra and ^{13}C -NMR spectra of compound 3f into the revised supplementary information part.

Reviewer #3

The content presented in the manuscript is from a well-done research. It presents a tandem research from the authors about their exciting recent articles on the high stereoselectivity outcome involving chiral VQM intermediates. In particular, they report here a novel synthetic approach to specific atropisomers in a very high enantioselective fashion. They went further and identified one of its novel compounds with good biological activity, in addition to considering its physicochemical properties. Hopefully, in the near future, they can identify a synthetically VQM-based drug in advanced preclinical stages, which demonstrate good in vivo efficacy, reasonable in vivo PK, and good safety profile.

Following are a few comments on the manuscript.

We greatly appreciate the reviewer's comments on our work.

Comment 1:

Chiral transformations from (S)-1k (Fig. 3b) need more rigorous stereochemical assignments. The chiral atropisomeric determinations would not be assigned as analogy from the transformations using organocatalysts (like others throughout the manuscript), but as a result of the intramolecular VQM chiral induction by the stereogenic sulfinamide group. Ideally, stereochemical assignment of any compound from the list of **1ka** to **1kd** would be obtained from X-ray crystallography. However, physical property comparison of anyone of these enantiomers with their corresponding counterparts could be enough. For example, although putative enantiomers **2k** and **1kb** are shown to have identical nmr spectra in SI (including traces of impurities), additional physical data would be desirable, such as demonstration that regular HPLC (not chiral HPLC) coelution of a made mixture of **1kb** [obtained from (S)-1k] and (R)-**2k** would result in one single peak as evidence that they are not diastereomers. Also, in a different determination using chiral HPLC, spiking the corresponding racemic mixture with **1kb** would show increase of one of the two peaks (rather than showing three peaks).

Response:

We thank the reviewer for raising this important point and we took careful investigation of the stereochemical assignments of **1kb** from **1k** by following your kind suggestion. First, we have tried to grow up a single crystal of products **1ka-1kd** for X-ray crystallography analysis, unfortunately none of these single crystals were obtained. Instead, we have tested the physical property of **1kb** and comparison with their corresponding counterparts by both of chiral HPLC and normal HPLC, and we have added this part of investigation into our revised Supplementary Information. The results are showing below:

As showing in Fig R7, product **1kb** obtained from **1k** were demonstrated to be the enantiomer of product **2k** by adding one of them into the racemic **2k** in chiral HPLC analysis. Meanwhile, the retention time of **1kb**, **2k** and rac-**2k** were identical in normal HPLC spectrums, indicating that they are not diastereomers.

Chiral HPLC analysis of 2k and 1kb:

Chiralcel OD-H, Hexane/*i*-PrOH = 85:15, flow rate = 1.0 mL/min, wave length = 254 nm

rac-2k

2k

2k + rac-2k

1kb

1kb + rac-2k

Fig R7. The chiral HPLC analysis of 1kb and 2k

Regular HPLC analysis of 2k and 1kb:

SHIMADZU(LC-20AD), C18(4.6*250 mm, 5 μ m),
Methanol/H₂O = 75:25, flow rate = 1.0 mL/min, wave
length = 254 nm.

rac-2k, $t_R = 7.050$ min

2k, $t_R = 7.045$ min

1kb, $t_R = 7.033$ min

2k + rac-2k, $t_R = 7.035$ min

1kb + rac-2k, $t_R = 7.036$ min

Fig R8. The regular HPLC analysis of 1kb and 2k

Comment 2:

All sufinamide examples have a gem-dimethyl group at the propargylic carbon, have you done the reaction when the propargylic carbon does not have substituents (i.e, a CH₂), or when it is attached to only one methyl group, or is part of a ring, e.g., cyclopropyl, cyclopentyl, etc.?

Response:

We thank the very interesting suggestion of this reviewer and we have examined a variety of substrates as you suggested. First, we have prepared the substrate in which the propargylic carbon does not have substituent and subjected it into our standard reaction conditions, as a result, product **2u** can be successfully formed but the enantioselectivity was very poor (Fif R9, 40% ee). Second, we also investigated the reaction by using the substrate with only one methyl group attached to propargylic carbon, while the enantioselectivity was not satisfactory at current stage. Instead of gem-dimethyl group at the propargylic carbon, we also tried gem-diethyl group, however, the corresponding substrate provides the target isothiazol with relatively low yield and poor enantioselectivity. Furthermore, we also investigated the substrate with substitutions at the propargylic carbon in a part of a ring, substrate **1x** with a cyclohexyl substitution were examined as well, disappointingly, the enantioselectivity was too poor at current stage. We have summarized these results into the end of Paragraph 4 on Page 3 in our revised manuscript.

Fig R9. The investigation of gem-disubstituted substrates

Comment 3:

For the gram scale preparation of (*aS*, *S*)-2k and (*S*)-1k, please include the recrystallization yields for both compounds.

Response:

We have added the recrystallization yields of products **2k** and **1k** in our revised manuscript.

Comment 4:

Please either specify or not the stereogenic assignment when writing the ID of chiral compounds throughout the manuscript. For example, always when referring to compound 1k, write either (*S*)-1k or just 1k.

Response:

We thank the reviewer for point out this problem and we have revised the ID of chiral compounds throughout the manuscript.

Comment 5:

Page 8, bottom. It is indicated that 4s IC₅₀ is 28.15 uM, however, I think, Fig. 5b shows 4s IC₅₀ is under 20 uM. Could you please correct it?

Response:

We have corrected this typo error in Fig.5b of our revised manuscript.

Comment 6:

Page 8, bottom. It is common that IC₅₀ potency/activity of enantiomers is different, so I would suggest replacing the statement “the two enantiomers were not exactly equivalent in bioactivity, for 4x was slightly more potent than its enantiomer counterpart (IC₅₀ of 2.57 μM versus 3.6 μM)” with “the two enantiomers were similar in bioactivity, where 4x was slightly more potent than its enantiomer counterpart (IC₅₀ of 2.57 μM versus 3.6 μM, Table S4)”.

Response:

We thank very much for the kind suggestion of the reviewer, we have replace the statement with “the two enantiomers were similar in bioactivity, where 4x was slightly more potent than its enantiomer counterpart (IC₅₀ of 2.57 μM versus 3.6 μM, Table S7)”.

Comment 7:

Page 9. I think the phrase “performed with compound 4x (Fig. 5b)” should be replaced with “performed with compound 4x (Fig. 5c)”.

Response:

We have corrected this typo error in our manuscript. The phrase “performed with compound 4x (Fig. 5b)” has been replaced with “performed with compound 4x (Fig. 5c)”.

Comment 8:

Page 11. I think the phrase “compared to the non-treated group (Fig. 6c)” should be replaced with “compared to the non-treated group (Fig. 6b)”.

Response:

The phrase “compared to the non-treated group (Fig. 6c)” has been replaced with “compared to the non-treated group (Fig. 6b)”.

Comment 9:

Page 11. I think the phrase “77.4% (8 μ M), respectively (Fig. 6d)” should be replaced with “77.4% (8 μ M), respectively (Fig. 6c)”.

Response:

We have replaced the phrase “77.4% (8 μ M), respectively (Fig. 6d)” with “77.4% (8 μ M), respectively (Fig. 6c)”.

Comment 10:

Page 11, top. Please replace “or replaced with methanol group” with “or replaced with hydroxymethyl group”.

Response:

We have replaced the phrase “or replaced with methanol group” with “or replaced with hydroxymethyl group”.

Comment 11:

Page 12, upper paragraph. Please replace “Moreover, take the axially chiral” with “Moreover, taking the axially chiral”.

Response:

We have replaced the phrase “Moreover, take the axially chiral” with “Moreover, taking the axially chiral”.

Comment 12:

Please consider replacing "de novo" with “novel” throughout the paper. To me, “novel” sounds more accurate.

Response:

We have replaced the phrase “de novo” with “novel”.

Comment 13:

Please replace "1, 2 azoles" with "1,2-azoles".

Response:

We have replaced the phrase “1, 2 azoles” with “1,2-azoles”.

Comment 14:

Page 8. I would suggest to replace “with alkyl groups gave rise to higher enantiomeric purity” with “with alkyl groups gave rise to slightly higher enantiomeric purity on average”.

Response:

We have replaced the phrase “with alkyl groups gave rise to higher enantiomeric purity” with “with alkyl groups gave rise to slightly higher enantiomeric purity on average”.

Comment 15:

Page 8. Please replace “were assigned as analogues” with “were assigned by analogy”. This suggested correction occurs twice on the same page.

Response:

We have replaced the phrase “were assigned as analogues” with “were assigned by analogy” all over the manuscript.

REVIEWERS' COMMENTS

Reviewer #1 (Remarks to the Author):

I appreciate the modifications done by Qin and Yan, especially in the determination of enantiomerization barriers and investigations concerning the extension of the scope. This study represents significant progress and novelty to justify its publication in Nature Communications in its form.

Reviewer #3 (Remarks to the Author):

Thanks to the authors for considering all comments. I support publication in Nature Communications after addressing the following final minor considerations.

1. SI page S37. For compound 1v, what is the stereochemical assignment for the propargylic carbon? If 1v is a mixture of diastereomers with one half "R" and one half "S" at the propargylic carbon, it should be clearly established. The same applies for 2v.
2. SI page S42. Please add the recrystallization yields of products 2k and 1k within the experimental procedure in the SI section (in parenthesis together with the ee results).
3. SI page S43. Please replace "The physical property of 1kb and 2k by both of chiral HPLC and normal HPLC As showing in Fig S1, product 1kb obtained from 1k were demonstrated to be the enantiomer of product 2k by adding one of them into the racemic 2k in chiral HPLC analysis." with "Physical property comparison of 1kb and 2k by both chiral HPLC and normal HPLC
As shown in Fig S1, 1kb, obtained from 1k, and 2k were demonstrated to be enantiomers of each other by mixing either one with rac-2k and running chiral HPLC analysis."
4. SI page S44. Please replace "The retention time of 1kb, 2k and rac-2k were identical in normal HPLC spectrums, indicating that they are not diastereomers." with "Retention times of 1kb, 2k and rac-2k were identical in normal HPLC chromatograms, suggesting that they are not diastereomers."
5. Manuscript page 5. Please replace "Nevertheless, substrates 1 with other gem-disubstituted groups (dihydro (2u), diethyl (2w), cyclohexyl (2x)) attached to the propargylic carbon provides the target isothiazole derivatives with extremely low yields and poor enantioselectivities under the standard reaction conditions (For details please see Supplementary Discussion)" with "However, substrate 1 with other gem-disubstituted groups [e.g, dihydro (2u), diethyl (2w), cyclohexyl (2x)] attached to the propargylic carbon provides the target isothiazole derivatives with extremely low yields and poor enantioselectivities under the standard reaction conditions (See Supplementary Information for details)"

Point-by-point responses to referees for manuscript

Reviewer #1 (Remarks to the Author):

I appreciate the modifications done by Qin and Yan, especially in the determination of enantiomerization barriers and investigations concerning the extension of the scope. This study represents significant progress and novelty to justify its publication in Nature Communications in its form.

Response: We greatly appreciate the reviewer's comments on our work.

Reviewer #3 (Remarks to the Author):

Comment 1:

SI page S37. For compound 1v, what is the stereochemical assignment for the propargylic carbon? If 1v is a mixture of diastereomers with one half "R" and one half "S" at the propargylic carbon, it should be clearly established. The same applies for 2v.

Response:

We thank the reviewer for pointing this out. We have made some studies to identify the stereochemical assignment for the propargylic carbon of compound 1v and 2v. Firstly, we have tried to prepare the single crystals of both products for X-ray analysis, unfortunately, we did not obtain the qualified single crystals for both of products 1v and 2v. In the meantime, we have done some literature search to address this issue, to our delight, the known literateurs were sufficient to help us for assigning the absolute configuration of the propargylic carbon in products 1v and 2v.

First, as showing in Fig 1a, Ellman and co-workers reported the first example of NaBH₄ reduction of enantiopure (*R*)-*N*-*tert*-butanesulfinyl ketimines,^[1] the *N*-*tert*-butanesulfinyl R-branched amines were obtained in good yield and excellent diastereoselectivities; when (*S*)-*N*-*tert*-butanesulfinyl ketimines acts as substrate in this reduction^{[2],[3]}, the *N*-*tert*-butanesulfinyl S-branched amines were acquired (Fig 1b)

Fig 1. The stereocontrol for the reduction of *N*-*tert*-butanesulfinyl ketimines by NaBH₄.

We synthesized substrate **rac-1v** by using literature methods (Fig 2).^{[1]-[6]} To a solution of a ketone

1 (1.0 equiv) and racemic (\pm)-*tert*-butanesulfinamide **2** (1.2 equiv) in THF (0.10 M) was added Ti(OEt)₄ (2.0 equiv) at room temperature. After stirring for the indicated time at 75 °C, the reaction mixture was cooled to room temperature and quenched by adding brine. After stirring for 30 min at the same temperature, insoluble materials were filtered through celite and washed with EtOAc thoroughly. The filtrate was concentrated and then purified by column chromatography on silica gel to afford the corresponding racemic *N-tert*-butanesulfinyl ketimines (\pm)**3**.^[4]

Racemic *N-tert*-butanesulfinyl ketimines (\pm)**3** (1.0 equiv) was dissolved in THF (0.10 M) and cooled to 0 °C. To the mixture was then added NaBH₄ (3.1 equiv), and the resulting solution was warmed to room temperature over a 3 h period. Only one new point was generated after TLC monitored of ketimines (\pm)**3** consumption, this indicates that the diastereoselectivity of the product **rac-4** could be well controlled during the reduction of *N-tert*-butanesulfinyl ketimine (\pm) **3** by NaBH₄, which was consistent with the literature reports.^{[1]-[3],[5],[6]} And then, **rac-1v** was obtained from **rac-4** after the removal of MOM group by *p*-TsOH.

Fig.2 Our procedure for the synthesis of **rac-1v**.

After obtained the substrate **rac-1v**, we obtained the chiral product **2v** (80% ee, d.r. > 20:1, 45% yield) and one enantiomer of **rac-1v** (74% ee, d.r. > 20:1, 48% yield) through kinetic resolution under our standard conditions (Fig 3). According to the absolute configuration of chiral (*aS*, *S*)-**2e** and (*S*)-**1e** (the absolute configuration of both product were assigned through X-ray single crystal diffraction analysis) and the reaction mechanism, it is speculated that the conversion rate of (*Rs*, *R*)-**1v** to (*aS*, *Ss*, *R*)-**2v** was much higher than that of (*Ss*, *S*)-**1v**, so the absolute configuration of the product **2v** was deduced to be (*aS*, *Ss*, *R*) and the absolute configuration of the chiral **1v** was (*Ss*, *S*); at the same time, we have also updated the corresponding information in the Supplementary Information.

Fig.3 The asymmetric synthesis of compound 2v.

References:

- [1] G. Borg, D. Cogan, J. Ellman, *Tetrahedron Lett.* **1999**, 40, 6709-6712.
- [2] C. Malapit, D. Caldwell, I. Luvaga, J. Reeves, I. Volchkov, N. Gonnella, Z. Han, C. Busacca, A. Howell, C. Senanayake, *Angew. Chem. Int. Ed.* **2017**, 56, 6999-7002.
- [3] Y. Gui, S. Tian, *Org. Lett.* **2017**, 19, 1554-1557.
- [4] G. Liu, T. Owens, T. Tang, J. Ellman, *J. Org. Chem.* **1998**, 64, 1278-1284.
- [5] M. Robak, M. Herbage, J. Ellman, *Chem. Rev.* **2010**, 110, 3600-3740.
- [6] J. Colyer, J. Tedrow, T. Soukup, M. Faul, *J. Org. Chem.* **2006**, 71, 6859-6862.

Comment 2:

SI page S42. Please add the recrystallization yields of products 2k and 1k within the experimental procedure in the SI section (in parenthesis together with the ee results).

Response:

We thank the reviewer for problem and have added the recrystallization yields and the ee values of products 2k and 1k within the experimental procedure in the Supplementary Information section.

Comment 3:

SI page S43. Please replace “The physical property of 1kb and 2k by both of chiral HPLC and normal HPLC

As showing in Fig S1, product 1kb obtained from 1k were demonstrated to be the enantiomer of product 2k by adding one of them into the racemic 2k in chiral HPLC analysis.” with “Physical property comparison of 1kb and 2k by both chiral HPLC and normal HPLC

As shown in Fig S1, 1kb, obtained from 1k, and 2k were demonstrated to be enantiomers of each other by mixing either one with rac-2k and running chiral HPLC analysis.”

Response:

Thanks a lot for your kind suggestion, we have replaced “The physical property of 1kb and 2k by both of chiral HPLC and normal HPLC

As showing in Fig S1, product 1kb obtained from 1k were demonstrated to be the enantiomer of product 2k by adding one of them into the racemic 2k in chiral HPLC analysis.” with “Physical property comparison of 1kb and 2k by both chiral HPLC and normal HPLC.

As shown in Fig S1, 1kb, obtained from 1k, and 2k were demonstrated to be enantiomers of each other by mixing either one with rac-2k and running chiral HPLC analysis.”

Comment 4:

SI page S44. Please replace “The retention time of 1kb, 2k and rac-2k were identical in normal HPLC spectrums, indicating that they are not diastereomers.” with “Retention times of 1kb, 2k and rac-2k were identical in normal HPLC chromatograms, suggesting that they are not diastereomers.”

Response:

Thanks a lot for your kind suggestion, we have replaced “The retention time of 1kb, 2k and rac-2k were identical in normal HPLC spectrums, indicating that they are not diastereomers.” with “Retention times of 1kb, 2k and rac-2k were identical in normal HPLC chromatograms, suggesting that they are not diastereomers.”

Comment 5:

Manuscript page 5. Please replace “Nevertheless, substrates 1 with other gem-disubstituted groups (dihydro (2u), diethyl (2w), cyclohexyl (2x)) attached to the propargylic carbon provides the target isothiazole derivatives with extremely low yields and poor enantioselectivities under the standard reaction conditions (For details please see Supplementary Discussion)” with “However, substrate 1 with other gem-disubstituted groups [e.g, dihydro (2u), diethyl (2w), cyclohexyl (2x)] attached to the propargylic carbon provides the target isothiazole derivatives with extremely low yields and poor enantioselectivities under the standard reaction conditions (See Supplementary Information for details)”

Response:

Thanks a lot for your kind suggestion, we have replaced “Nevertheless, substrates 1 with other gem-disubstituted groups (dihydro (2u), diethyl (2w), cyclohexyl (2x)) attached to the propargylic carbon provides the target isothiazole derivatives with extremely low yields and poor enantioselectivities under the standard reaction conditions (For details please see Supplementary Discussion)” with “However, substrate 1 with other gem-disubstituted groups [e.g, dihydro (2u), diethyl (2w), cyclohexyl (2x)] attached to the propargylic carbon provides the target isothiazole derivatives with extremely low yields and poor enantioselectivities under the standard reaction conditions (See Supplementary Information for details)”